# Harm reduction strategies in acute care for people who use alcohol and/or drugs: A scoping review

Daniel Crowther[1], Janet Curran[1,2]*, Mari Somerville[1,2], Doug Sinclair[2], Lori Wozney[3], Shannon MacPhee[2], Annette Elliott Rose[2], Leah Boulos[4], Alexander Caudrella[5]

1 School of Nursing, Dalhousie University, Halifax, Nova Scotia, Canada, 2 Quality and Patient Safety, IWK Health, Halifax, Nova Scotia, Canada, 3 Mental Health and Addictions Program, Nova Scotia Health Authority, Halifax, Nova Scotia, Canada, 4 The Maritime Strategy for Patient Oriented Research SUPPORT Unit, Halifax, NS, Canada, 5 Mental Health and Addictions Service, St Michael's Hospital, Toronto, Ontario, Canada

* jacurran@dal.ca

**Data Availability Statement:** This review relies on data from publicly available published literature.

## Abstract

### Background

People who use alcohol and/or drugs (PWUAD) are at higher risk of infectious disease, experiencing stigma, and recurrent hospitalization. Further, they have a higher likelihood of death once hospitalized when compared to people who do not use drugs and/or alcohol. The use of harm reduction strategies within acute care settings has shown promise in alleviating some of the harms experienced by PWUAD. This review aimed to identify and synthesize evidence related to the implementation of harm reduction strategies in acute care settings.

### Methods

A scoping review investigating harm reduction strategies implemented in acute care settings for PWUAD was conducted. A search strategy developed by a JBI-trained specialist was used to search five databases (Medline, Embase, CINAHL, PsychInfo and Scopus). Screening of titles, abstracts and full texts, and data extraction was done in duplicate by two independent reviewers. Discrepancies were resolved by consensus or with a third reviewer. Results were reported narratively and in tables. Both patients and healthcare decision makers contributing to the development of the protocol, article screening, synthesis and feedback of results, and the identification of gaps in the literature.

### Findings

The database search identified 14,580 titles, with 59 studies included in this review. A variety of intervention modalities including pharmacological, decision support, safer consumption, early overdose detection and turning a blind eye were identified. Reported outcome measures related to safer use, managed use, and conditions of use. Reported barriers and enablers to implementation related to system and organizational factors, patient-provider communication, and patient and provider perspectives.

Included articles can be identified through the reference list.

**Funding:** This project is funded by the Strategy for Patient-Oriented Research Evidence Alliance in partnership with the Canadian Institutes of Health Research (https://sporevidencealliance.ca/ Grant number: Q80-21). The grant was awarded to JC. The funders had no role in study design, data collection and analysis, decision to publish, or preparation of the manuscript.

**Competing interests:** The authors have declared that no competing interests exist.

## Conclusion

This review outlines the types of alcohol and/or drug harm reduction strategies, which have been evaluated and/or implemented in acute care settings, the type of outcome measures used in these evaluations and summarizes key barriers and enablers to implementation. This review has the potential to serve as a resource for future harm reduction evaluation and implementation efforts in the context of acute care settings.

## Introduction

Substance use is a global health concern, with drug and/or alcohol misuse contributing to over 5% of the global burden of disease [1]. People who use alcohol and/or drugs (PWUAD) are at an increased risk of infectious disease, recurring hospitalizations and death [2, 3]. A recent cross-Canadian study highlighted the increasing potency, unpredictability, and poor quality of unregulated substances placing PWUAD at a greater risk of serious health outcomes, such as poisoning [4]. While better management of substance use is recognized as a priority by national and international governing bodies, there are challenges in supporting PWUAD in the health care system. The complex social and health care needs of PWUAD create significant barriers in accessing care [5]. A 2010 report from the World Health Organization found that mental health care services, which provide substance use supports, are underutilized by PWUAD [1]. In addition to poor access to substance use support across the health and social care system, PWUAD often face stigma when seeking treatment for substance use disorders from health care providers [6]. Improved health system supports are urgently needed to ensure all PWUAD receive adequate care.

Harm reduction is an approach that emphasizes working with people where they are at, rather than focusing solely on drug and alcohol abstinence [7]. Harm reduction strategies are a promising approach for health care providers and health systems to improve the care of PWUAD. Harm reduction strategies such as safe injection sites, needle exchange programs and methadone maintenance treatment have led to reduced deaths from overdose [8], a decrease in human immunodeficiency virus (HIV) infections [9], and fewer hospitalizations [3]. Additionally, the recent decision by the U.S. Department of Health and Human Services' to remove the X-waiver requirement for the use of medication-assisted treatment for people who use opioids highlight that harm reduction strategies are increasingly being viewed as valid and necessary approaches to care [10]. Despite advances in our understanding of the effectiveness of harm reduction strategies, greater public buy in, and the need for enhanced access to health care services for PWUAD, there are gaps in how and when PWUAD receive care.

While the majority of harm reduction work for PWUAD has been conducted in the community, evidence suggests that hospitals represent an ideal setting for implementing harm reduction strategies [11]. Hospitals are an ideal point of care for PWUAD, with rates of hospital admission and emergency department (ED) utilization higher for PWUAD than the general population [12]. In a qualitative study, PWUAD reported that hospital-based harm reduction strategies would enhance patient-centred care by promoting a culturally safe environment, ensuring timely access to care and prioritizing substance use symptoms [2]. However, health care providers repeatedly report lack of training as being a barrier to providing quality care to PWUAD in the hospital setting [13]. Further, there is little known about how harm reduction strategies are implemented in the inpatient and ED setting, making it challenging to design effective and acceptable interventions for this population. Therefore, the aim of this study is to

map the evidence related to the implementation of harm reduction strategies in inpatient and ED settings for PWUAD. This study will answer four research questions:

1. What harm reduction strategies have been evaluated to help alleviate negative health outcomes associated with substance use within inpatient settings and EDs?

2. What are the commonly reported outcome measures used to evaluate harm reduction strategies and their implementation in these settings?

3. How are harm reduction strategies implemented in inpatient settings and EDs?

4. What are the reported barriers and enablers to their implementation?

## Methods

### Study design

This review followed the JBI methodology for scoping reviews [14] and was registered with Open Science Framework (Registration DOI: 10.17605/OSF.IO/P7BHN). This review utilized an integrated knowledge translation (iKT) approach [15]. The methods of the review were previously reported [16] and are briefly summarized below.

### Search strategy

The search strategy was designed by a JBI-trained information specialist (LB), in collaboration with the study team and the principle knowledge user (AC), and was peer reviewed by a second research librarian (S1 Table). Five electronic databases were searched for articles: Medline, Embase, CINAHL, PsychInfo and Scopus. An additional search of the grey literature was completed following the systematic approach of Godin et al. [17].

### Inclusion criteria

The participants, concept and context framework [14] was used to characterize the research question. Participants included either PWUAD who were accessing acute care settings for any health condition, or participants who provided care to PWUAD in acute care settings. Participants of any race, age and gender were considered for inclusion. Any studies evaluating interventions or implementation of interventions with the goal of reducing harms related to alcohol and/or substance use were considered. Studies utilizing patient reported outcome measures, patient reported experience measures and health outcome measures we considered for inclusion. Studies which took place in acute care settings (e.g., inpatient settings, emergency departments) were also considered. Outpatient services, primary care, community settings, long term inpatient settings (e.g., nursing homes, prisons) were excluded. Opinion papers, commentaries, newspaper articles were not included. Evidence syntheses were not included, however, the reference lists of any relevant evidence synthesis were searched for applicable articles. Grey literature sources (e.g., policy documents, unpublished program evaluations) were searched and assessed for eligibility. Articles were only included if they reported in English and the full text was available.

### Screening approach

Articles retrieved from the search were managed using Covidence [18]. Following de-duplication, articles were independently screened by two reviewers, starting with titles and abstracts, and followed by full text papers. Any disagreements between reviewers were resolved through discussion or by consultation with the research team.

## Data extraction

Data were extracted from each included study using a predetermined data extraction form. The data extraction form was pilot tested with six team members using one of the included studies. The team met to review any discrepancies in data extraction and to refine the data extraction tool. The data from each study were independently extracted in duplicate. The team met regularly to discuss any concerns related to the data extraction process until data extraction was complete.

Extracted data included characteristic and demographic details such as country, year of publication, study design, objective, participant sample, and setting characteristics. Intervention details included the type and length of intervention, the population targeted by the intervention, what type of drug use the intervention addressed and the reported outcomes measures. Implementation details included whether provider training, sustainability, quality and performance, cost, communication or participant compensation was mentioned. Additionally, data related to reported barriers or enablers to implementation was extracted.

## Data synthesis and presentation

Extracted data were synthesized into four major sections (population characteristics, intervention characteristics, characteristics of outcome measures and implementation characteristics) using tables, figures and narrative description. The reporting and presentation of this review followed the Preferred Reporting Items for Systematic Reviews and Meta-analyses for Scoping Reviews (PRISMA-ScR) (S2 Table) [19].

To further elucidate the harm reduction principles present in each intervention approach, the interventions were coded by Hawk et al.'s [20] six harm reduction principles for healthcare settings. These include: humanism, pragmatism, individualism, autonomy, incrementalism and accountability without termination. Interventions which contained pharmacological approaches were coded under pragmatism. Interventions which provided multiple services, tailored their services to meet patient needs and allowed shared decision making in terms of goal setting were coded to individualism. Interventions which supported patients during readmission, provided follow-up and ongoing care and aimed to "meet patients where they are" were coded to incrementalism. Interventions which provided information and/or referrals to additional services and care and allowed patients to make decisions related to their care were coded under autonomy. Interventions which provided education to patients and caregivers related to risks associated with continued drug use and overdose were coded to accountability without termination. Finally, interventions which enacted system changes and emphasized empathy, support, understanding and compassion were coded under humanism.

Barriers and enablers to implementation were organized based on the following pre-defined categories: system and organizational factors, provider-patient communication, patient perceptions and provider perceptions. System and organizational factors encompass barriers/enablers related to law and policy, funding and resources, and structural and environmental factors such availability of hospital space and the physical location of a hospital. Provider-patient communication encompass barriers/enablers related to communication between providers and patients. Finally, patient perceptions and provider perceptions encompass barrers/enablers related to how patient and providers perceive the harm reduction strategy.

## Patient, decision maker and community organization engagement

Patients and healthcare decision makers were engaged throughout this review to: a) ensure that our research questions aligned with priorities relevant to all partners; b) allow partners with lived experiences of harm reduction programs to contextualize our findings and; c)

inform the identification of key gaps in the literature that may have been overlooked without their engagement.

In addition to patients and healthcare decision makers, the preliminary findings of this review were also shared with members of a community organization which provides harm reduction services as part of their community health promotion and disease prevention mandate. Individuals from this organization were given the opportunity to provide feedback and comment on the findings, drawing on their own experiences of receiving, providing and advocating for care. This feedback was used to interpret review findings and integrated into the discussion. Reporting of engagement adhered to the Guidance for Reporting Involvement of Patients and the Public revised short form (GRIPP2-SF) checklist (S3 Table) [21].

### Critical appraisal

All studies were critically appraised by two independent reviewers using the Mixed Methods Appraisal Tool (MMAT) [22]. The MMAT can be applied to a range of study designs and is therefore useful for scoping reviews. Based on the quality of the report, each article received a score ranging from 0 to 5. Differences in scoring were resolved by consensus or a third reviewer. Scores are listed in Table 1.

## Results

### Included studies

The database search returned 36,264 records, with 14,580 titles remaining after deduplication. No additional relevant articles were found through the grey literature search. A total of 549 full texts were assessed for eligibility, with 59 studies meeting the inclusion criteria for this review. Our patient partner screened 982 titles and abstracts. An overview of the screening process can be found in **Fig 1** (Preferred Reporting Items for Systematic and Meta Analyses (PRISMA) Flow Diagram).

### General characteristics of included studies

Included studies utilized a range of study designs with retrospective chart reviews (n = 14) [23–36] being the most common, followed by randomized controlled trials (n = 10) [37–46], cross-sectional designs (n = 10) [47–56], case reports (n = 9) [57–65], qualitative studies (n = 5) [66–70], mixed methods studies (n = 5) [71–75], prospective cohort design (n = 2) [76, 77], prospective observational design (n = 1) [78], pre-/post-test design (n = 1) [79], interrupted time series design (n = 1) [80], and quasi experimental: non-equivalent group design (n = 1) [81]. The majority of included articles were conducted in the United States (n = 39) [24–42, 44, 45, 47–50, 52–54, 61, 63, 66, 70–73, 75, 77, 80, 81], followed by Canada (n = 14) [23, 51, 55–59, 62, 64, 65, 67, 68, 74, 79], Australia (n = 3) [46, 69, 78], Netherlands (n = 2) [43, 60], and Sweden (n = 1) [76]. The majority of articles were published between 2019–2021 (n = 39) [23–38, 42, 45, 48–50, 52–56, 58, 59, 61, 63, 66–68, 70, 72–74, 78, 79], while the remaining were published between 1999 and 2018 (n = 20) [39–41, 43, 44, 46, 47, 51, 57, 60, 62, 64, 65, 69, 71, 75–77, 80, 81] (Table 1).

The reported sample size across studies ranged from 1 to 30,263. The majority of studies targeted patient populations alone (n = 46) [23–28, 30, 32–42, 44–52, 55, 56, 58–60, 62–68, 74–79, 81], followed by both patients and health care providers (HCPs) (n = 8) [31, 43, 54, 61, 69–71, 80], patients and caregivers (n = 3) [29, 57, 73] and HCPs alone (n = 2) [53, 72].

### Intervention characteristics

Thirty-two of the included articles reported delivering interventions in EDs [25, 27, 28, 30–34, 37, 40, 41, 44, 46, 47, 49, 51, 53–55, 61, 63, 66, 69, 71–76, 78, 80, 81], while twenty-two studies

**Table 1. Summary of included articles.**

| Author, Year | Country | Setting | Study design | Study Objectives | MMAT |
|---|---|---|---|---|---|
| **Banta-Green et al., 2019** [37] | USA | ED | Randomized Controlled Trial (RCT) | Tested an intervention for opioid users at elevated risk for overdose to determine the impact on participants' subsequent opioid overdoses, ED visits and hospitalisations. | **** |
| **Boora et al., 2021** [79] | Canada | Inpatient—MHA | Pre-/Post-test Design (Chart Review) | Assess the feasibility of a new seamless care transition and to assess its affect on our outcome measure of wait time to first outpatient MHT assessment and readmission rate to hospital. | *** |
| **Boudreaux et al., 2015** [81] | USA | ED | Quasi Experimental: Non-equivalent groups | Describes initial development, functionality, acceptability, and, overall feasibility of a new telehealth SBIRT delivery model. | **** |
| **Brooks et al., 2019** [23] | Canada | Inpatient—General Medicine | Retrospective Cohort Design (Chart Review) | Identify the proportion of intakes in which patients were offered and accepted syringes and analyse demographic characteristics that predict offering and accepting syringes. | **** |
| **Byrne et al., 2020** [38] | USA | Inpatient—General Medicine | RCT | Compare the effectiveness of an inpatient link to recovery coaching services to the current standard of care. | *** |
| **Christian et al., 2021** [24] | USA | Inpatient—General Medicine | Retrospective Cohort Design (Chart Review) | Evaluate a hospitalist-led interprofessional program created to identify hospitalized patients with OUD, initiate buprenorphine in the inpatient setting, and provide bridge prescription and access to outpatient treatment programs | *** |
| **Cushman et al., 2016** [39] | USA | Inpatient—General Medicine | RCT | Determine if inpatient buprenorphine initiation and linkage to outpatient buprenorphine reduce injection opiate users' frequency of injection opiate use. | * |
| **D'Onofrio et al., 2008** [41] | USA | ED | RCT | Assess the efficacy of an emergency practitioner performed intervention in reducing alcohol consumption and negative consequences | ***** |
| **D'Onofrio et al., 2012** [40] | USA | ED | RCT | Evaluate the efficacy of the Brief Negotiation Interview | ***** |
| **Désy and Perhats, 2008** [71] | USA | ED | Mixed Methods Design (Chart Review and Semi-structured Interviews) | Conduct the SBIRT process with eligible patients. Explore barriers and enablers to implementing and maintaining SBIRT | ** |
| **Devries et al., 2019** [25] | USA | ED | Retrospective Cohort Design (Chart Review) | Assess whether routine screening increases identification of candidates for take-home naloxone, increases naloxone prescribing to those who need it, and identify which screening tool questions were highest yield for identifying naloxone candidates. | **** |
| **Dong et al., 2020** [58] | Canada | Inpatient—General Medicine | Case Report | Report the implementation of a supervised consumption services in acute care setting for inpatients. | *** |
| **Dwyer et al., 2015** [47] | USA | ED | Cross Sectional Design | Evaluate the feasibility of an ED-based overdose prevention and intervention program, and describe overdose risk knowledge, opioid use, overdose, and overdose response actions among ED patients | **** |
| **Eswaran et al., 2020** [66] | USA | ED | Qualitative | Summarize the individual and collective experiences in implementing take-home naloxone programs within the Chicago area. | - |
| **Eswaran et al., 2020** [48] | USA | ED & Inpatient | Cross Sectional Design (Chart Review) | Describe the development of an ED-based take-home naloxone program | **** |
| **Gerdtz et al., 2020** [78] | Australia | ED | Prospective Observational Design | To determine the prevalence of amphetamine-type stimulant among patients admitted to the ED. Explore the referral outcomes for those who tested positive for, or who self-reported amphetamine-type stimulant use. | ***** |
| **Gryczynski et al., 2021** [42] | USA | ED & Inpatient | RCT | Examine the effectiveness of Navigation Services to Avoid Rehospitalization (NavSTAR) in promoting engagement in care and reducing acute care use | ***** |
| **Holland et al., 2020** [72] | USA | ED | Mixed Methods Design (Chart Review and Semi-structured Interviews) | To assess feasibility and implementation of a centered computerized clinical decision support system | ** |

*(Continued)*

**Table 1.** (Continued)

| Author, Year | Country | Setting | Study design | Study Objectives | MMAT |
|---|---|---|---|---|---|
| **Hurt et al., 2020** [49] | USA | ED | Cross Sectional Design | Study ED patients with active opioid misuse regarding their prior knowledge of naloxone and to identify barriers to obtaining and using naloxone kits | **** |
| **Jakubowski et al., 2019** [50] | USA | Inpatient—General Medicine | Cross Sectional Design (Chart Review) | To evaluate the processes of a naloxone program and the patients reached | *** |
| **Johnson et al., 2016** [57] | Canada | Inpatient—General Medicine | Case Report | Describes the implementation of a naloxone distribution program | *** |
| **Johnson et al., 2020** [73] | USA | ED | Mixed Methods (Retrospective Chart Review + Directed Content Analysis) | Describes an implementation-focused process evaluation of a brief intervention for substance users | ***** |
| **Joosten et al., 2009** [43] | Netherlands | Inpatient—MHA | RCT | Examine the effect of shared-decision making on patients and clinicians perceptions of the therapeutic alliance | ** |
| **Kestler et al., 2017** [51] | Canada | ED | Cross Sectional Design | Determine patient acceptance of ED-based naloxone program and examined factors related to acceptance. | ***** |
| **Kestler et al., 2019** [74] | Canada | ED | Mixed Methods (Cross Sectional Survey using Closed-and Open-ended questions) | Determine why intravenous drug users accept or decline take home naloxone | *** |
| **Kirby et al., 2021** [26] | USA | Inpatient—Rehab | Retrospective Cohort Design (Chart Review) | Evaluate the effect of this new program content (the opioid series) on rates of MAT within 30 days of completion of inpatient rehabilitation with a diagnosis of OUD. Evaluate the effect of MAT and completion of the opioid series was investigated in relation to rates of OUD-related ED visits and/or hospitalization admission within 1 year after rehabilitation program completion | ***** |
| **Kosteniuk et al., 2021** [67] | Canada | ED & Inpatient | Qualitative | Examine key factors that shape patients' decisions to attend or not attend a novel supervised consumption service embedded within a large, urban acute care hospital. | ***** |
| **Ladak et al., 2021** [59] | Canada | Inpatient—MHA | Case Report | We present the case of a 40-year-old patient with OUD using illicit fentanyl, heroin, and oxycodone preoperatively and admitted for an elective liver resection for steroid-induced hepatoma. | *** |
| **Liebling et al., 2021** [52] | USA | ED & Inpatient | Cross Sectional Design (Chart Review) | Describe the implementation of hospital-based peer recovery support services for substance use disorder | ***** |
| **Monti et al., 1999** [44] | USA | ED | RCT | Evaluate the use of brief motivational interview to reduce alcohol related consequences and use among adolescents treated in the ER following an alcohol related event | *** |
| **Moore et al., 2021** [53] | USA | ED | Cross Sectional Design (Chart Review) | Evaluate the feasibility of the Point of Care naloxone protocol, and report the rate of obtainment in comparison to previously published references in the literature | *** |
| **Mullennix et al., 2020** [54] | USA | ED | Cross Sectional Design | Describe the implementation of a clinical nurse specialist–led emergency department overdose education and naloxone distribution program. | **** |
| **Nordqvist et al., 2005** [76] | Sweden | ED | Prospective Cohort Design | Evaluate whether screening without one-to-one feedback and screening with simply written advice are sufficient to initiate a self-regulation process concerning risky drinking among emergency care patients | **** |
| **O'Brien et al., 2019** [55] | Canada | ED | Cross Sectional Design (Chart Review) | Describe a take home naloxone program offered to patients presenting in ED with opioid overdose | ***** |
| **Papp et al., 2019** [27] | USA | ED | Retrospective Cohort Design (Chart Review) | Determine the impact naloxone rescue kits had on repeat opioid overdose related ED visits, hospitalization, and death | **** |
| **Parappilly et al., 2020** [68] | Canada | Inpatient—General Medicine | Qualitative | Describe the experience of patients with severe alcohol use disorder on a managed alcohol program while admitted to an acute care hospital | ***** |
| **Prach et al., 2019** [28] | USA | ED | Retrospective Cohort Design (Chart Review) | Implement and evaluate an inpatient naloxone program in a community teaching hospital | ** |

*(Continued)*

**Table 1.** (Continued)

| Author, Year | Country | Setting | Study design | Study Objectives | MMAT |
|---|---|---|---|---|---|
| **Ray et al., 2020** [29] | USA | Inpatient—General Medicine | Retrospective Cohort Design (Chart Review) | Evaluate the feasibility of implementing a comprehensive, multidisciplinary treatment approach in the management of patients with opioid use disorder admitted to cardiovascular surgery for medical or surgical treatment | **** |
| **Samuels et al., 2019** [30] | USA | ED | Retrospective Cohort Design (Chart Review) | Determine whether ED naloxone distribution and recovery coach consultation improves frequency and timeliness of linkage to opioid treatment, reduces recurrent opioid overdose, and reduces incidence of opioid overdose death | ***** |
| **Samuels et al., 2018** [75] | USA | ED | Mixed Methods (Retrospective Chart Review And Open-ended survey questions) | Measure the sustainability of an ED take-home naloxone and peer recovery coach consultation program for ED patients at risk of opioid overdose. | **** |
| **Samuels et al., 2019** [70] | USA | Inpatient—General Medicine | Qualitative | Assess the facilitators and barriers of implementing treatment standards for care of adult patients with opioid use disorder | * |
| **Samuels et al., 2021** [31] | USA | ED | Retrospective Cohort Design (Chart Review) | Evaluated the influence of Levels of Care policy implementation on offering and receipt of take home naloxone, ED behavioral counselling, and referral to treatment following and ED visit for opioid overdose | *** |
| **Schipper et al., 2018** [60] | Netherlands | Inpatient—Critical Care | Case Report | Explore the effects of substituting street cannabis with a low THC medicinal cannabis variant in patients with a psychotic disorder and comorbid CUD in terms of acceptance, craving and severity of psychotic symptoms | *** |
| **Schreyer et al., 2020** [61] | USA | ED | Case Report | Present the case of a patient found after a presumed opioid overdose in our ED | *** |
| **Sise et al., 2005** [80] | USA | ED | Interrupted Time Series Design | Describe the implementation of a screening, brief intervention and referral program at a urban teaching hospital trauma center | *** |
| **Snyder et al., 2021** [32] | USA | ED | Retrospective Cohort Design (Chart Review) | Evaluated the implementation of low-threshold ED buprenorphine treatment at 52 hospitals participating in the CA Bridge Program | ***** |
| **Stein et al., 2021** [45] | USA | Inpatient—General Medicine | RCT | Test whether a brief intervention in skin cleaning would result in greater reductions in follow up ED visits or hospitalization rates when compared with a usual care condition | **** |
| **Tait et al., 2004** [46] | Australia | ED | RCT | Evaluate the effectiveness of a brief intervention enhanced by a consistent support person in facilitating attendance for substance use treatment following a hospital alcohol or other drug presentation. | ** |
| **Thompson et al., 2020** [33] | USA | ED | Retrospective Cohort Design (Chart Review) | Determine impact of a substance use consultant on length of stay and hazard ratio for a routine hospital discharge | **** |
| **Townsend, 2021** [34] | USA | ED | Retrospective Cohort Design (Chart Review) | Investigated the utilization of Peer Recovery Specialists in an ED setting | **** |
| **Train et al., 2020** [35] | USA | Inpatient—General Medicine | Retrospective Cohort Design (Chart Review) | A quality improvement (QI) project was aimed to increase the prescription of naloxone kits at patient discharge | *** |
| **Van Heukelom et al., 2019** [56] | Canada | Inpatient—General Medicine | Cross Sectional Design | A quality improvement study was undertaken to explore the perception of nurses in caring for patients on hospital based managed alcohol program | *** |
| **Wakeman et al., 2017** [77] | USA | Inpatient—General Medicine | Prospective Cohort Design | Determine whether inpatient addiction consultation improves substance use outcomes 1 month after discharge. | *** |
| **Weiland et al., 2008** [69] | Australia | ED | Qualitative | Evaluate emergency staff attitudes in performing alcohol screening and delivering opportunistic brief intervention; and to document process issues associated with the introduction of routine clinician-initiated opportunistic screening and training and administration of brief intervention. | *** |

(*Continued*)

**Table 1.** (Continued)

| Author, Year | Country | Setting | Study design | Study Objectives | MMAT |
|---|---|---|---|---|---|
| **Weinrib et al., 2017** [62] | Canada | ED & Inpatient | Case Report | This report describes the postsurgical management of a patient with complex chronic pain and high-dose opioid dependence after urgent surgery. | *** |
| **Weinstein et al., 2020** [36] | USA | Inpatient—General Medicine | Retrospective Cohort Design (Chart Review) | Evaluate whether an addiction consult is associated with acute care utilization | **** |
| **Welch et al., 2019** [63] | USA | ED | Case Report | Describe the Relay program (a harm reduction strategy) | **** |
| **Young et al., 2002** [64] | Canada | Inpatient–Critical Care | Case Report | A case illustrating the potential harms of discontinuation of outpatient methadone upon admission to an intensive care unit | *** |
| **Young et al., 2018** [65] | Canada | Inpatient–Rehab | Case Report | A case report of a dedicated team using harm reduction principles that was formed and trained to work with PWUAD in a spinal cord program | *** |

MMAT:

* (1) = one out of five criterion met

** (2) = two out of five criterion met

*** (3) = three out of five criterion met

**** (4) = four out of five criterion met

***** (5) = five out of five criterion met.

[23, 24, 26, 29, 35, 36, 38, 39, 43, 45, 50, 56–60, 64, 65, 68, 70, 77, 79] took place in inpatient settings and five studies [42, 48, 52, 62, 67] included both ED and inpatient settings. Of the articles that delivered interventions in inpatient settings, specific departments included general

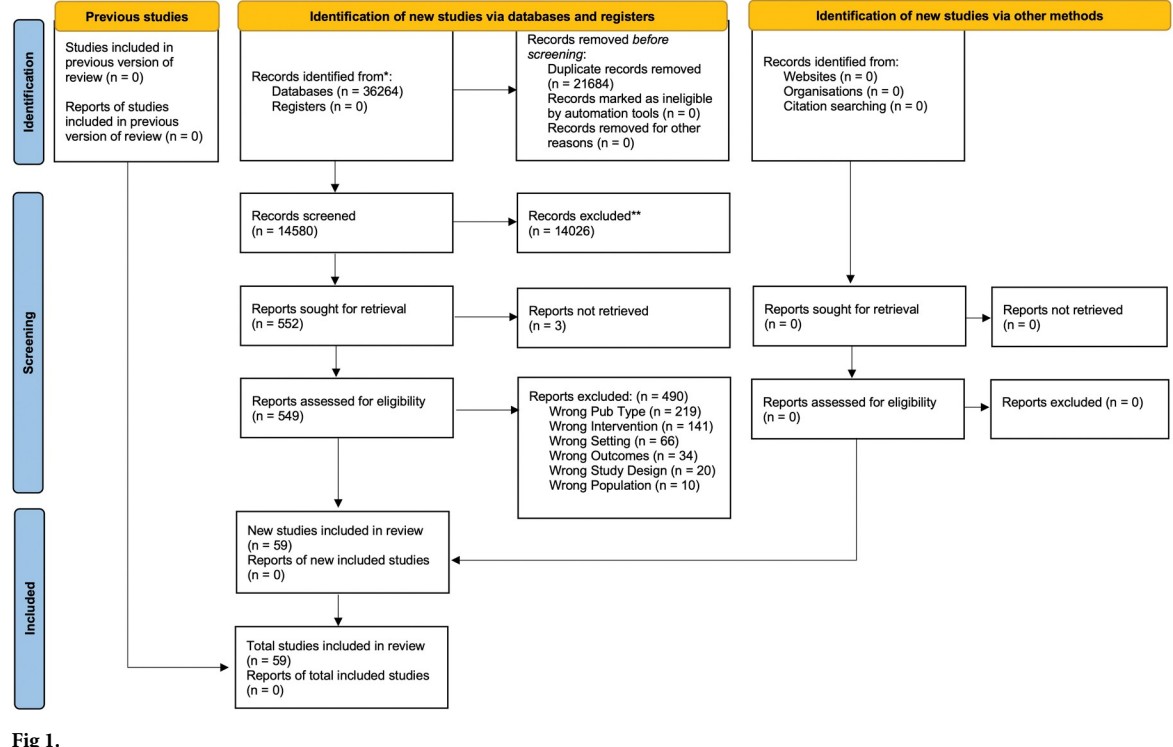

**Fig 1.**

medicine (n = 15) [23, 24, 29, 35, 36, 38, 39, 45, 50, 56–58, 68, 70, 77], critical care (n = 2) [60, 65], psychiatric, mental health or pain programs (n = 3) [43, 59, 79], and rehabilitation programs (n = 2) [26, 64].

Interventions targeted several types of drug use, including opioid use (n = 31) [24–32, 35, 37, 39, 47–55, 57, 61–63, 65, 66, 70, 72, 74, 75], alcohol use (n = 8) [40, 41, 44, 56, 68, 69, 71, 76], amphetamine use (n = 1) [78] or cannabis use (n = 1) [60]. Interventions which were not targeted to address a specific drug or targeted poly-drug use were categorized under substance use and represented 18 of the included articles [23, 33, 34, 36, 38, 42, 43, 45, 46, 58, 59, 64, 67, 73, 77, 79–81]. Interventions targeting alcohol and opioid use were predominantly set in the ED, whereas interventions targeting substance use were predominantly set in inpatient settings.

Interventions which targeted patients and caregivers did not report adapting intervention components based on the presence of the caregiver, only that caregivers were welcome to participate in the intervention alongside patients. Interventions which targeted HCP populations primarily utilized education or clinical pathway decision support materials. Only two articles [31, 70] utilized policy change as a strategy, and those interventions were targeted at both patient and HCP populations.

This review identified interventions which used several different modalities to reduce harms associated with drug and/or alcohol use, these included pharmacological (n = 4) [56, 60, 65, 68], decision support (n = 22) [29, 34, 36, 38, 40–46, 52, 53, 69, 71–73, 76, 78–81], safer consumption (n = 3) [23, 58, 67], early overdose detection (n = 1) [61] and turning a blind eye (n = 1) [64]. Twenty-eight articles [24–28, 30–33, 35, 37, 39, 47–51, 54, 55, 57, 59, 62, 63, 66, 70, 74, 75, 77] combined pharmacological and decision support approaches, with most of these being related to naloxone education and distribution, and primarily occurring in the ED. Other pharmacological strategies included the use of buprenorphine, methadone, cannabinoid derivatives and alcohol.

Decision support covered a range of intervention approaches including education, training, counselling, referral, and clinical decision supports. The use of brief interventions was a common approach (n = 10) [40, 41, 44, 46, 69, 71, 73, 76, 80, 81] within the decision support category. These interventions exclusively occurred in the emergency departments and addressed either alcohol use (n = 6) [40, 41, 44, 69, 76, 81] or substance use (n = 4) [46, 73, 80, 81]. The use of peer recovery supports (n = 9) [30, 31, 34, 38, 52, 59, 63, 70, 75] and specialized teams (n = 4) [24, 36, 77, 79] were also common and targeted opioid and substance use exclusively.

Safer consumption approaches included supervised consumption sites (n = 2) [58, 67] and needle distribution programs (n = 1) [23]. Only one article [61] reported using early overdose protection and the intervention was in the form of an emergency department bathroom sensor that detected when individuals remained immobile for a period of time. Only one article [64] reported utilizing the turning a blind eye approach, which involved ignoring patient drug use at a park across the street from the facility. Table 1 presents a full overview of intervention characteristics.

All of the included interventions utilized at least two of Hawk et al.'s harm reduction principles, with pragmatism being the most common (n = 34) [23–28, 30–33, 35, 37–39, 47–51, 53–58, 60, 63, 66, 68, 70, 74–76, 78], followed by individualism (n = 33) [24, 25, 29–31, 33, 36, 38, 40–44, 46, 52, 57–59, 62–65, 67, 70–75, 77, 79–81], incrementalism (n = 24) [24, 28, 31, 34, 36, 38–43, 45, 53, 57–59, 61, 62, 65, 67, 71, 73, 77, 79], autonomy (n = 19) [27, 29, 30, 35, 37, 43, 44, 46, 47, 50, 54, 59, 64, 66, 67, 70, 75, 80, 81], accountability without termination (n = 19) [23, 24, 26, 30, 37, 39–41, 44, 46–48, 50, 55, 63, 64, 68, 76, 80] and humanism (n = 14) [28, 34, 36, 40, 41, 44, 45, 52, 58, 59, 61, 65, 72, 79] (Table 2).

**Table 2. Characteristics of interventions by setting.**

| Author, Year | Population Targeted | Type of Drug Use | Intervention Approaches | Intervention Content | Humanism | Pragmatism | Individualism | Autonomy | Incrementalism | Accountability |
|---|---|---|---|---|---|---|---|---|---|---|
| **Ambulatory: Emergency Department** | | | | | | | | | | |
| **Banta-Green et al., 2019** [37] | Patients | Opioid Use | Pharmacological; Decision Support | Opioid overdose education and naloxone kit distribution program | | ✓ | | ✓ | | ✓ |
| **Boudreaux et al., 2015** [81] | Patients | Substance Use | Decision Support | Screen to identify appropriate intervention, brief intervention to increase insight and awareness regarding substance use and motivation toward behavioral change. Referral to treatment | | | ✓ | ✓ | | |
| **D'Onofrio et al., 2008** [41] | Patients | Alcohol Use | Decision Support | Brief intervention to increase insight and awareness regarding substance use and motivation toward behavioral change, negotiated drinking goals, provided pamphlet of harm risks | ✓ | | ✓ | | ✓ | ✓ |
| **D'Onofrio et al., 2012** [40] | Patients | Alcohol Use | Decision Support | Brief intervention to increase insight and awareness regarding substance use and motivation toward behavioral change, negotiated drinking goals | ✓ | | ✓ | | ✓ | ✓ |
| **Désy and Perhats, 2008** [71] | Patients & HCP | Alcohol Use | Decision Support | Screen to identify appropriate intervention, brief intervention to increase insight and awareness regarding substance use and motivation toward behavioral change. Referral to treatment | | | ✓ | | ✓ | |
| **Devries et al., 2019** [25] | Patients | Opioid Use | Pharmacological; Decision Support | A protocol for HCPs to teach and distribute Naloxone kits to patients | | ✓ | ✓ | | | |
| **Dwyer et al., 2015** [47] | Patients | Opioid Use | Pharmacological; Decision Support | Opioid overdose education and naloxone kit distribution program | | ✓ | | ✓ | | ✓ |
| **Eswaran et al., 2020** [48] | Patients | Opioid Use | Pharmacological; Decision Support | Opioid overdose education and naloxone kit distribution program | | ✓ | | | | ✓ |
| **Gerdtz et al., 2020** [78] | Patients | Amphetamine Use | Decision Support | Screened for illicit substances, provided with a harm reduction brochure, and offered a referral. | | ✓ | | | | |
| **Holland et al., 2020** [72] | HCP | Opioid Use | Decision Support | Clinical decision support guidance for drug therapy using a computerized system | ✓ | | ✓ | | | |
| **Hurt et al., 2020** [49] | Patients | Opioid Use | Pharmacological; Decision Support | Opioid overdose education and naloxone kit distribution program | | ✓ | | | | |

*(Continued)*

**Table 2.** (Continued)

| Author, Year | Population Targeted | Type of Drug Use | Intervention Approaches | Intervention Content | Humanism | Pragmatism | Individualism | Autonomy | Incrementalism | Accountability |
|---|---|---|---|---|---|---|---|---|---|---|
| Johnson et al., 2020 [73] | Patients & Caregivers | Substance Use | Decision Support | Screen to identify appropriate intervention, brief intervention to increase insight and awareness regarding substance use and motivation toward behavioral change. Referral to treatment | | | ✓ | | ✓ | |
| Kestler et al., 2017 [51] | Patients | Opioid Use | Pharmacological; Decision Support | Opioid overdose education and naloxone kit distribution program | | ✓ | | | | |
| Kestler et al., 2019 [74] | Patients | Opioid Use | Pharmacological; Decision Support | Opioid overdose education and naloxone kit distribution program | | ✓ | ✓ | | | |
| Monti et al., 1999 [44] | Patients | Alcohol Use | Decision Support | Brief intervention to increase insight and awareness regarding substance use and motivation toward behavioral change, negotiated drinking goals | ✓ | | ✓ | ✓ | | ✓ |
| Moore et al., 2021 [53] | HCP | Opioid Use | Decision Support | Protocol for opioid overdose education and naloxone kit distribution program | | ✓ | | | ✓ | |
| Mullennix et al., 2020 [54] | Patients & HCP | Opioid Use | Pharmacological; Decision Support | Opioid overdose education and naloxone kit distribution program | | ✓ | | ✓ | | |
| Nordqvist et al., 2005 [76] | Patients | Alcohol Use | Decision Support | Brief intervention to increase insight and awareness regarding substance use and motivation toward behavioral change. | | ✓ | | | | ✓ |
| O'Brien et al., 2019 [55] | Patients | Opioid Use | Pharmacological; Decision Support | Opioid overdose education and naloxone distribution program | | ✓ | | | | ✓ |
| Papp et al., 2019 [27] | Patients | Opioid Use | Pharmacological; Decision Support | Opioid overdose education and naloxone kit distribution program | | ✓ | ✓ | ✓ | | |
| Prach et al., 2019 [28] | Patients | Opioid Use | Pharmacological; Decision Support | Opioid overdose education and naloxone kit distribution program | ✓ | ✓ | | | ✓ | |
| Samuels et al., 2018 [75] | Patients | Opioid Use | Pharmacological; Decision Support | Opioid overdose education and naloxone kit distribution program, Recovery Coach consultation and referral support | | | ✓ | ✓ | | |
| Samuels et al., 2019 [30] | Patients | Opioid Use | Pharmacological; Decision Support | Opioid overdose education and naloxone kit distribution program, Recovery Coach consultation and referral support | | ✓ | ✓ | ✓ | | ✓ |

(Continued)

Table 2. (Continued)

| Author, Year | Population Targeted | Type of Drug Use | Intervention Approaches | Intervention Content | Humanism | Pragmatism | Individualism | Autonomy | Incrementalism | Accountability |
|---|---|---|---|---|---|---|---|---|---|---|
| **Samuels et al., 2021** [31] | Patients & HCP | Opioid Use | Pharmacological; Decision Support | Policy level changes. For Patients: education on safe opioid storage and disposal, substance use disorder screening, referral to treatment on discharge, peer recovery consultation, urine drug testing for fentanyl and fentanyl analogs, naloxone distribution. For HCP: mandated reporting of suspected opioid overdoses, staff training. Ensuring that facilities can meet these needs |  | ✓ | ✓ |  | ✓ |  |
| **Schreyer et al., 2020** [61] | Patients & HCP | Opioid Use | Early Overdose Detection | Reverse motion detector in ED Waiting Room Washroom | ✓ |  |  |  | ✓ |  |
| **Sise et al., 2005** [80] | Patients & HCP | Substance Use | Decision Support | Screen to identify appropriate intervention, brief intervention to increase insight and awareness regarding substance use and motivation toward behavioral change. Referral to treatment |  |  | ✓ | ✓ |  | ✓ |
| **Snyder et al., 2021** [32] | Patients | Opioid Use | Pharmacological; Decision Support | Low-threshold buprenorphine treatment approach, active patient navigation from ED care to outpatient addiction treatment, and harm reduction interventions in line with an overdose education and naloxone distribution methodology. |  | ✓ |  |  |  |  |
| **Tait et al., 2004** [46] | Patients | Substance Use | Decision Support | Brief intervention to increase insight and awareness regarding substance use and motivation toward behavioral change. Referral to treatment |  |  | ✓ | ✓ |  | ✓ |
| **Thompson et al., 2020** [33] | Patients | Substance Use | Pharmacological; Decision Support | Medication treatment initiation, motivational interviewing, Naloxone training, discharge planning, and linkages to resources |  | ✓ | ✓ |  |  |  |
| **Townsend, 2021** [34] | Patients | Substance Use | Decision Support | A peer recovery specialist consulted with patients, provided support for referral and linkage to treatment | ✓ |  |  |  | ✓ |  |

*(Continued)*

Table 2. (Continued)

| Author, Year | Population Targeted | Type of Drug Use | Intervention Approaches | Intervention Content | Humanism | Pragmatism | Individualism | Autonomy | Incrementalism | Accountability |
|---|---|---|---|---|---|---|---|---|---|---|
| Weiland et al., 2008 [69] | Patients & HCP | Alcohol Use | Decision Support | Brief intervention to increase insight and awareness regarding substance use and motivation toward behavioral change. | | | | | | |
| Welch et al., 2019 [63] | Patients | Opioid Use | Pharmacological; Decision Support | Peer navigation and support, opioid overdose education and naloxone distribution program, referral, and linkage to treatment | | ✓ | ✓ | | | ✓ |
| **Inpatient: General Medicine** | | | | | | | | | | |
| Brooks et al., 2019 [23] | Patients | Substance Use | Safer Consumption | Safe Syringe Distribution | | ✓ | | | | ✓ |
| Byrne et al., 2020 [38] | Patients | Substance Use | Decision Support | Recovery coach | | ✓ | ✓ | | ✓ | |
| Christian et al., 2021 [24] | Patients | Opioid Use | Pharmacological; Decision Support | Buprenorphine was initiated in-hospital by nurse; Psychospiritual care and counseling is offered by chaplain and social worker; A follow-up appointment is made for outpatient care in the community with a partner OBOT team following discharge | | ✓ | ✓ | | ✓ | ✓ |
| Cushman et al., 2016 [39] | Patients | Opioid Use | Pharmacological; Decision Support | Received buprenorphine/naloxone while in hospital and then at discharge, linked to a primary care buprenorphine clinic. | | ✓ | | | ✓ | ✓ |
| Dong et al., 2020 [58] | Patients | Substance Use | Safer Consumption | Supervised consumption service | ✓ | ✓ | ✓ | | ✓ | |
| Jakubowski et al., 2019 [50] | Patients | Opioid Use | Pharmacological; Decision Support | Opioid overdose education and naloxone kit distribution program | | ✓ | | ✓ | | ✓ |
| Johnson et al., 2016 [57] | Patients & Caregivers | Opioid Use | Pharmacological; Decision Support | Opioid overdose education and naloxone kit distribution program | | ✓ | ✓ | | ✓ | |
| Parappilly et al., 2020 [68] | Patients | Alcohol Use | Pharmacological | Alcohol is prescribed by the Addiction Medicine Consult using a customizable order set | | ✓ | | | | ✓ |

(Continued)

Table 2. (Continued)

| Author, Year | Population Targeted | Type of Drug Use | Intervention Approaches | Intervention Content | Humanism | Pragmatism | Individualism | Autonomy | Incrementalism | Accountability |
|---|---|---|---|---|---|---|---|---|---|---|
| **Ray et al., 2020** [29] | Patients & Caregivers | Opioid Use | Decision Support | The inpatient algorithm included assessment and evaluation, pre/post op pain management, psychotherapy, and maintenance. The discharge algorithm includes the notification of psychiatry/addiction medicine, choice between IV antibiotics, psychotherapy, treatment referral, inpatient/residential/intensive outpatient program | | | ✓ | ✓ | | |
| **Samuels et al., 2019** [70] | Patients & HCP | Opioid Use | Pharmacological; Decision Support | Policy level changes. For Patients: education on safe opioid storage and disposal, substance use disorder screening, referral to treatment on discharge, peer recovery consultation, urine drug testing for fentanyl and fentanyl analogs, naloxone distribution. For HCP: mandated reporting of suspected opioid overdoses, staff training, and ensuring that facilities are capable of meeting these needs | | ✓ | ✓ | ✓ | | |
| **Stein et al., 2021** [45] | Patients | Substance Use | Decision Support | A skin hygiene education and skills-training behavioral intervention | ✓ | | | | ✓ | |
| **Train et al., 2020** [35] | Patients | Opioid Use | Pharmacological; Decision Support | Opioid overdose education and naloxone kit distribution program | | ✓ | | ✓ | | |
| **Van Heukelom et al., 2019** [56] | Patients | Alcohol Use | Pharmacological | Small, measured doses of beverage alcohol (e.g., beer, wine, or spirits) with the intention of preventing withdrawal, reducing consumption of non-beverage alcohol (e.g., hand sanitizer or mouthwash) and ensuring that alcohol consumption occurs in a safe environment. | | ✓ | | | | |
| **Wakeman et al., 2017** [77] | Patients | Substance Use | Pharmacological; Decision Support | A multidisciplinary specialty team offering pharmacotherapy initiation, motivational counseling, treatment planning, and direct linkage to ongoing addiction treatment. | | | ✓ | | ✓ | |

(*Continued*)

Table 2. (Continued)

| Author, Year | Population Targeted | Type of Drug Use | Intervention Approaches | Intervention Content | Humanism | Pragmatism | Individualism | Autonomy | Incrementalism | Accountability |
|---|---|---|---|---|---|---|---|---|---|---|
| **Weinstein et al., 2020** [36] | Patients | Substance Use | Decision Support | A multi-disciplinary team counsels about treatment options, coaches and collaborates with inpatient providers, initiate evidence-based medications for addiction and bridges patients to long-term outpatient treatment. | ✓ | | ✓ | | ✓ | |
| **Inpatient: Psychiatric, Mental Health or Pain Team** | | | | | | | | | | |
| **Boora et al., 2021** [79] | Patients | Substance Use | Decision Support | A referral pathway and a referral form to be used by inpatient teams when a patient requires follow-up care post-discharge; | ✓ | | ✓ | | ✓ | |
| **Joosten et al., 2009** [43] | Patients & HCP | Substance Use | Decision Support | Shared Decision Making. Both patient and clinician completed the Goals of Treatment questionnaire and discussed the results to reach a treatment agreement | | | ✓ | ✓ | ✓ | |
| **Ladak et al., 2021** [59] | Patients | Substance Use | Pharmacological; Decision Support | Peer Recovery support; coaching, mentoring, and or education by individuals with lived experience of recovery | ✓ | | ✓ | ✓ | ✓ | |
| **Inpatient: Rehabilitation** | | | | | | | | | | |
| **Kirby et al., 2021** [26] | Patients | Opioid Use | Pharmacological; Decision Support | Consists of four 1-hour group education sessions for participants. Psychologists and social workers lead 3 sessions, respectively focusing on motivational interviewing, triggers and coping skills, and coping with pain. A clinical pharmacy specialist leads a fourth session, providing education regarding MAT for OUD and the role of naloxone rescue kits for overdose. | | ✓ | | | | ✓ |
| **Young et al., 2002** [64] | Patients | Substance Use | Decision Support; Turning a blind-eye | The counselor, along with abstinence-based approaches, utilizes motivational interviewing rational recovery techniques and development of readiness for change–taking an individualized approach to all clients in the spinal cord program. A blind-eye is turned to drug use in an adjacent park. | | | ✓ | ✓ | | ✓ |
| **Inpatient: Critical Care** | | | | | | | | | | |

(*Continued*)

**Table 2.** (Continued)

| Author, Year | Population Targeted | Type of Drug Use | Intervention Approaches | Intervention Content | Humanism | Pragmatism | Individualism | Autonomy | Incrementalism | Accountability |
|---|---|---|---|---|---|---|---|---|---|---|
| **Schipper et al., 2018** [60] | Patients | Cannabis Use | Pharmacological | Replacing the use of street cannabis with high tetrahydrocannabinol and low cannabidiol levels by medical cannabis variants with low tetrahydrocannabinol and high cannabidiol | ✓ | ✓ | | | | |
| **Young et al., 2018** [65] | Patients | Opioid Use | Pharmacological | Methadone administration to prevent the harm related to methadone treatment interruption | ✓ | | ✓ | | ✓ | |
| **Combination: Emergency Department & Inpatient** | | | | | | | | | | |
| **Eswaran et al., 2020** [48] | Patients | Opioid Use | Pharmacological; Decision Support | Opioid overdose education and naloxone kit distribution program | | ✓ | | ✓ | | |
| **Gryczynski et al., 2021** [42] | Patients | Substance Use | Decision Support | NavSTAR (harm reduction strategy), used proactive case management, advocacy, service linkage, and motivational support to resolve internal and external barriers to care and address SUD, medical, and basic needs for 3 months after discharge | | | ✓ | | ✓ | |
| **Kosteniuk et al., 2021** [67] | Patients | Substance Use | Safer Consumption | Supervised consumption service | | | ✓ | ✓ | ✓ | |
| **Liebling et al., 2021** [52] | Patients | Opioid Use | Decision Support | Peer Recovery support; coaching, mentoring, and or education by individuals with lived experience of recovery. | ✓ | | ✓ | | | |
| **Weinrib et al., 2017** [62] | Patients | Opioid Use | Pharmacological; Decision Support | Postsurgical opioid weaning is supported, and pain management is optimized using multimodal approaches, including behavioral strategies | | | ✓ | | ✓ | |

## Characteristics of outcome measures

Reported outcome measures were organized into four categories, based on those defined by G. Alan Marlatt (1996) and the National Harm Reduction Coalition: [82, 83] abstinence, safer use, managed use, and conditions of use and use itself (Table 3). Safer use was comprised of outcome measures related to pharmacological distribution and/or acceptance, syringe acceptance, treatment implementation and presence of safe consumption site. Safer use outcome measures were used most frequently in ED settings. Managed use was comprised of outcome measures related to referral to and/or acceptance of care, satisfaction and/or experience of care and HCP follow-up. Conditions of use and use itself included measures related to mortality, readmission rates, leaving against medical advice (AMA), adverse events, length of stay and frequency of use of drugs and/or alcohol. One article, Schreyer et al., 2020 [61], did not report outcome measures related to these categories, however they did report outcome measures related to implementation. No studies reported on abstinence.

## Characteristics of intervention implementation

Twenty-one articles reported at least one study aim related to measuring intervention implementation, with two of these described as quality improvement studies. The most commonly

**Table 3. Characteristics of outcome measures by setting.**

| | | Ambulatory: Emergency Department (n =) | Inpatient: General Medicine (n =) | Inpatient: Psychiatric, Mental Health or Pain Team (n =) | Inpatient: Rehab (n =) | Inpatient: Critical Care(n =) | Combination: Emergency Department & Inpatient (n =) |
|---|---|---|---|---|---|---|---|
| Safer Use | Pharmacological Distribution/ Acceptance | 11 [25, 31, 32, 48, 49, 53, 55, 58, 63, 72, 74] | 3 [35, 57, 70] | 0 | 0 | 0 | 1 [66] |
| | Syringe Acceptance | 0 | 1 [23] | 0 | 0 | 0 | 0 |
| | Treatment Implementation | 1 [30] | 0 | 0 | 0 | 0 | 0 |
| | Safe Consumption Site | 0 | 1 [58] | 0 | 0 | 0 | 0 |
| Managed Use | Referral/ Acceptance of Care | 7 [31, 32, 34, 46, 72, 73, 78] | 3 [24, 29, 38] | 0 | 1 [26] | 0 | 3 [52, 62, 67] |
| | Satisfaction/ Experience of Care | 6 [51, 68, 69, 71, 80, 81] | 1 [56] | 2 [43, 59] | 1 [65] | 2 [60, 64] | 1 [62] |
| | HCP Follow-up | 2 [40, 81] | 2 [28, 45] | 1 [43] | 0 | 0 | 2 [52, 62] |
| Conditions of Use and Use Itself | Mortality | 5 [27, 33, 37, 54, 75] | 0 | 0 | 1 [26] | 0 | 1 [42] |
| | Readmission Rate | 4 [33, 34, 37, 75] | 6 [28, 29, 36, 39, 45, 77] | 1 [79] | 1 [26] | 1 [64] | 2 [42, 66] |
| | Leaving AMA | 2 [33, 55] | 2 [36, 50] | 1 [59] | 1 [65] | 1 [64] | 0 |
| | Adverse Events | 3 [41, 44, 63] | 0 | 0 | 0 | 1 [60] | 0 |
| | Length of Stay | 2 [27, 33] | 4 [29, 35, 36, 38] | 0 | 1 [65] | 0 | 0 |
| | Drug or Alcohol Use | 6 [27, 40, 41, 46, 47, 76] | 2 [39, 77] | 0 | 0 | 0 | 0 |

reported factor related to implementation was provider training (n = 33) [24, 26–28, 31–33, 35, 37–41, 43–45, 47, 48, 50, 52–54, 57, 61, 64, 69–71, 73, 75, 78, 80, 81], followed by sustainability (n = 21) [26, 30, 32, 33, 35, 39, 40, 47, 48, 53, 54, 61–64, 67, 70, 72, 73, 80, 81], providing a honorarium/credit for participants (n = 15) [32, 37–41, 43–45, 51, 67–69, 74, 77], quality and performance (n = 15) [29, 31, 33–35, 40–42, 44, 54, 56, 61, 70, 73, 81], cost (n = 14) [32, 34, 35, 42, 47, 48, 53, 54, 61, 63, 66, 70, 80, 81], and communication and marketing (n = 6) [28, 54, 56, 66, 67, 72]. Less than half of the articles reported implementation factors related to cost, sustainability, communication and marketing, quality and performance, and honorarium/credit for participants.

## Reported barriers and enablers to implementation

Thirty-eight (64%) of the included articles reported at least one barrier and/or enabler to implementation (Table 4). The most commonly reported barriers and enablers were related to system and organizational factors. Factors related to patient-provider communication, patient perspectives and provider perspectives were also reported.

## Quality appraisal

Overall the quality of the included studies was moderate to high with 2% (n = 1) [66] scoring 0, 3% (n = 2) [39, 70] scoring 1, 8% (n = 5) [28, 43, 46, 71, 72] scoring 2, 36% (n = 21) [24, 31, 35, 38, 44, 50, 53, 56–62, 64, 65, 69, 74, 77, 79, 80] scoring 3, 29% (n = 17) [23, 25, 27, 29, 33, 34, 36, 37, 45, 47–49, 54, 63, 75, 76, 81] scoring 4, and 22% (n = 13) [26, 30, 32, 40–42, 51, 52, 55, 67, 68, 73, 78] scoring 5 (Table 1).

## Partner reflections on gaps in the literature

The following provides additional reflections from our patient and decision maker community partners on gaps identified through our review (Table 5). This section aims to share further contextual details that may be useful to researchers and policy makers. Consultations with partners took place after the preliminary data chart was developed.

## Discussion

Overall, this review identified a diverse range of harm reduction strategies, which had been evaluated and implemented in different types of acute health care settings. Strategies were primarily implemented in the ED, followed by a range of inpatient settings such as general medicine, rehabilitation, critical care, and mental health or pain teams. The identified outcome measures used for strategy evaluation were related to safe use, managed use and use itself. Finally, the included studies did not report detailed implementation strategies or the use of frameworks to guide implementation. Factors related to implementation were inconsistently reported across articles, however a range of barriers and enablers were reported, albeit primarily related to system and organizational level factors.

 Our findings suggest that there is a growing interest in the implementation of harm reduction strategies within acute care settings, as publications on this subject have dramatically increased within the last three years. While EDs have typically not considered such programs to be part of their mandate, calls for the use of harm reduction strategies within EDs have begun to emerge [84–86]. This apparent willingness to implement such strategies in ED settings is promising for the success of future projects, particularly given that PWUD seeking health care are more likely to do so via EDs [87]. However, while there appears to be growing support within EDs, we identified a limited number of studies which were set in non-mental

**Table 4. Barriers and enablers to implementation.**

| System and Organizational Factors: Reported Barriers of Implementation |
| --- |
| • Time and labor a burden for HCPs [41, 66, 70, 71, 73] |
| • Inadequate training and support for HCPs [48, 62, 66, 69, 70, 70, 81] |
| • Information barrier due to absence of formal guidelines [66] |
| • Staffing issues [29, 30, 73] |
| • Part-time health educators not available outside working hours [80] |
| • Lack of resources [29, 56, 62] |
| • Costs associated with implementation [30, 50, 54, 62] |
| • The fast paced nature of the ED, an emphasis on throughput and the constraints of acute care in the ED [53, 66] |
| • Referrals were limited by access to specialists, and treatment services in the hospital and the surrounding community [71] |
| • Inadequate administrative support [71] |
| • Difficulty scheduling or locating patients for training. |
| • Lack of case management when addressing complex psychosocial illnesses [29] |
| • Insurance status [30, 66] |
| • Lack of organizational prioritization [70] |
| • Unengaged and remote hospital leadership [70] |
| • Poor interdepartmental communication, collaboration, and coordination [70] |
| • Poor community-hospital partnerships [70] |
| • Absence of local champion [70] |
| • Rural and community hospitals had structural barriers to follow-up treatments [29, 32] |
| • COVID 19 initiated closures and decreased intakes to addiction treatment programs creating barriers to care for treatment seeking patients [32] |
| • Screening impacted by patient trauma [80] |
| • Timely identification of patients [62, 69] |
| • No systematic approach to selecting clients for program and inability to select clients based on potential future alcohol/drug use [64] |
| • Phones not available in in all treatment areas [81] |
| • Adaptability of intervention [70] |
| • Trained patients forgetting naloxone kits at discharge; only patient who received training used their kits [49, 57] |
| • Covering the cost of and sourcing naloxone kits [54, 66, 70] |
| • Challenges related to documenting the prescribing, dispensing, and training of naloxone distribution [48, 53, 54, 57]. |
| • Inability to dispense naloxone to admitted patients because admitted patients are not permitted to receive discharge medication in the ED prior to transport into the hospital [53] |
| • Ambiguity on dispensing status [66] |
| • Ambiguity on labeling procedures [66] |
| • Ambiguity on legal liability [66] |
| • Obtaining X-waiver to prescribe BUP, and access and availability to MOUD in the community [24, 72] |
| • Limited outpatient availability of specialists for OUD [70] |
| • SCS was not designed or equipped to support supervised drug inhalation [67] |
| • SCS was only available to registered inpatients, leaving visitors to use drugs in hospital in unsafe areas [67] |
| • Lack of education and unclear guideline related to MAP [56] |
| **System and Organizational Factors: Reported Enablers of Implementation** |
| • Brief interventions which require no additional resources to implement [40, 76, 78] |
| • Securing funding [32, 48, 66] |
| • Multidisciplinary team involvement including pharmacists, nurses, and practitioners from various specialties [66] |
| • Community organization assistance [66, 70] |
| • Pharmacy engagement [66] |

*(Continued)*

**Table 4.** (Continued)

| |
|---|
| • Knowledge of state legislation [66] |
| • Collaboration with other surrounding hospitals [66] |
| • Support from hospital leadership; enabling program leaders to address administrative barriers in a timely manner [48, 53, 66, 70] |
| • Full time social workers with low caseloads and a modest discretionary fund [42] |
| • Communication and sharing with local champions and among colleagues [72] |
| • Access to sterile injection supplies and ability to safely disposal of used supplies [67] |
| • Partnerships between pharmacists and physicians [53] |
| • Policy adaptability to local context [70] |
| • Immediate intervention delivery [29, 69] |
| • Staff training [35, 70] |
| • Adopt protocols to work within the ED setting [32] |
| • Use of trained providers with an awareness of harm reduction principles [32, 53] |
| • Empowering clinicians to be "changemakers" reduced the stigma in the hospital and spurred adaptation of health care system to meet patients' needs [32] |
| • Outsourcing SBIR allowed the trauma service to continue to provide high-quality, consistent services [80] |
| • Public pressure to address opioid overdose crisis [70] |
| • Hospital to hospital competition [70] |
| • Public relations pressure to be seen as "taking action" [70] |
| • Regulatory requirements [70] |
| • Provision of state-sponsored training related to OUD medication [70] |
| • Interdepartmental collaboration [70] |
| • Electronic medical record order sets, provider reminders, custom forms, report generation [66, 70, 73] |
| • Dedicated staff for overdose reporting [70] |
| • Local expertise in addiction medicine [70] |
| • Provider knowledge about OUD and comfort with initiation of OUD medication [39] |
| • Technical assistance [70] |
| • Securing a supply of naloxone kits for dispensing at no cost to the patient [53, 66, 70] |
| • Leverage the availability of a nonphysician health care provider to provide naloxone education to patients [66] |
| • Protocols for screening patients for naloxone eligibility [53] |
| • Local champions in emergency medicine, social work, psychiatry [70] |
| • Outsourced bilingual health educators who provide direct patient services, record keeping, and information transfer to physicians and nurses [80] |
| **Provider-Patient Communication: Reported Barriers of Implementation** |
| • Difficulty maintaining contact with providers [64] |
| • Lack of privacy within the ED made patient discussions challenging [69, 71] |
| • Difficulty communicating with patient due to language, comprehension, intoxication, and/or altered conscious state [69] |
| • The clients are more likely than others to miss appointments and can be less motivated to reach their full physical potential [64] |
| • Discussing naloxone in a way that encourages patients to buy in [53] |
| • Providers concerned about offending their patients about the cost of naloxone [53] |
| • Patient dishonesty [69] |
| **Provider-Patient Communication: Reported Enablers of Implementation** |
| • Greater involvement of family members in care/treatment plans and addiction education with nurses and other health care providers [57] |
| • Open discussions with patient and family [62] |
| **Patient Perceptions: Reported Barriers of Implementation** |
| • Patient apprehension in discussing substance use with HCPs [67, 81] |

(*Continued*)

**Table 4.** (Continued)

| |
| --- |
| • Patient distrust of healthcare services [23, 67] |
| • Patient fear of being stigmatized by HCPs [67] |
| • Lack of trust that the hospital SCS would provide adequate protection from criminalization and surveillance [67] |
| **Patient Perceptions: Reported Enablers of Implementation** |
| • Increased patient awareness of high risk alcohol consumption [69] |
| **Provider Perceptions: Reported Barriers of Implementation** |
| • Doubt related to policy changes efficacy, screening/treatment efficacy and patient adherence [70, 71] |
| • The perception that psychosocial interventions are not the responsibility of ED HCPs [71] |
| • HCP resistance to changes in practice [54, 66] |
| • HCP resistance due to belief that they are encouraging drug use and increasing harms [56, 66] |
| • HCPs bias and stigma toward drug using patients [54, 70, 72] |
| • Lack of understanding of who would benefit from naloxone kits [53] |
| • Difficulty motivating HCPs to buy in [57, 69, 71] |
| • HCP opposition to harm reduction strategies and associated resource allocation [64] |
| **Provider Perceptions: Reported Enablers of Implementation** |
| • Belief in effectiveness of policy changes [70] |
| • Increased provider awareness of high risk alcohol consumption [69] |
| • Staff acceptance [73] |

health inpatient units. Training and harm reduction programs within these health service settings have the potential to be a valuable resource and reduce stigma. As such, further research exploring implementing programs in non-mental health inpatient settings are needed.

While this review identified a diverse range of harm reduction strategies, additional strategies currently being used in other settings were notably absent. Managing and providing nutrition for PWUAD [88], providing housing resources [89], drug checking technology to allow PWUAD to ascertain unknown chemicals in their street drugs [90, 91], and providing off label prescriptions (i.e., safe supply) to PWUAD [92] are all increasingly being considered as

**Table 5. Summary of partner reflections on gaps in the literature.**

| Summary of Partner Reflections on Gaps in the Literature |
| --- |
| • Using trauma-informed approaches to guide the design and implementation of harm reduction strategies should be considered. |
| • PWUAD can experience trauma through interacting with health services and they share these experiences with their peers. Creating and/or improving programs that allow for the reporting of neglect or abuse could help establish greater trust between PWUAD and providers and help improve issues related to accountability. |
| • Providing education related to safe use of all drugs, not just opioids should be considered when developing harm reduction strategies. |
| • Providing education to providers on the varied types of drug/alcohol patients may be using and providing education on best practices on how to interact with people who use drugs/alcohol should be considered. |
| • Treatment referrals are viewed as valuable, but these services can be located away from hospitals, and transit to and from these locations can be viewed as a significant barrier. |
| • When interacting with healthcare institutions PWUAD can feel like they must critically consider treatment recommendations before they follow them because they may feel that they know more about safe drug use than providers. |
| • PWUAD often have strong networks and information about safe use practices are often shared within communities. These networks have the potential to be a valuable information sharing resource. |
| • Future research should consider including PWUAD in the design and implementation of these programs. |
| • Strategies to deal with adverse events, such as allergic reactions to Naloxone or methadone, should be planned for and integrated into harm reduction programs. |

important harm reduction approaches, yet to our knowledge they have not been evaluated for use in EDs and/or inpatient settings. Additionally, our patient partners noted that providing education to PWUAD that improves their understanding of the health services they access could help set appropriate expectations on what kind of care they can expect to receive, potentially mitigating instances of leaving against medical advice and/or distrust of HCPs and health services. In the context of other healthcare services, educational strategies focused on health literacy have been shown to strengthen patient engagement [93] and improve patient health outcomes [94] and should therefore be considered as a potential avenue for additional harm reduction approaches.

Most outcome measures included in this review were designed to capture data related to program uptake, adherence and real world efficacy. As such, these outcome measures have the potential to inform the allocation of program resources and the tailoring of programs to specific contexts, making them valuable in informing program implementation, program evaluation and quality improvement projects. Of note, measurement of the satisfaction/experience of care was utilized in only 22% (n = 13) of the included studies. PWUAD dissatisfaction and poor experiences of care has been associated with stigma related to drug use, and this type of stigma has been identified as a factor in increased risk of leaving against medical advice and poor health outcomes [6, 95]. Additionally, abuse and suffering experienced as a result of accessing healthcare is more likely among stigmatized populations, is poorly understood and often goes unreported [96]. Measures of satisfaction or experience of care which are patient-oriented could be a valuable tool in widening our understanding of and managing these issues during program implementation and evaluation, potentially leading to improved patient outcomes.

While there were studies included in this review that reported implementation factors related to communication and marketing, cost, quality and performance, sustainability, and provider training, none of the studies utilized validated frameworks to inform their approach to implementation. Given the myriad of factors that can influence effective implementation [97] (e.g., rural/urban setting, available resources, level of personnel training, and patient/provider beliefs and attitudes) a greater emphasis on developing implementation strategies prior to implementation could help improve the effectiveness of approaches. This review also identified a range of barriers and enablers to implementation, most of which related to system and organizational level factors. Absent from these barriers and enablers was public awareness and opinion of harm reduction approaches. Negative public opinion of PWUAD and of harm reduction strategies can negatively impact the perceived value of certain strategies [98]. Media reporting of harm reduction services has the potential to reduce stigma against PWUAD and increase acceptance of harm reduction approaches [99]. In settings where HCP buy-in is a barrier, utilizing public messaging and information campaigns of the benefits of harm reduction services could help improve uptake by HCPs.

## Limitations

We included studies based on our definition of a harm reduction approach. Harm reduction is a broad philosophy encompassing a range of ideas and as such our definition may have limited the inclusion of some approaches. However, our team included a harm reduction specialist who was consulted throughout the process to ensure that we were comprehensive in our inclusion of approaches. This review sought to identify and describe the scope of the available literature and as such providing definitive recommendations related to the implementation and/or evaluation of specific programs was not possible. Further, this review is not a meta-analysis and as such, the generalizability of the results within each study was not assessed. Our

approach to data synthesis involved coding extracted data into validated frameworks (e.g., Hawk's harm reduction principles), this process involved some level of subjective interpretation. To mitigate these effects, coding was done independently by two reviewers who then resolved any discrepancies through consensus or consultation with the research team. Our partners were engaged throughout this review to help contextualize our findings and ensure that we maintained a patient-centered approach in our methodology and reporting. Partner reflections should only be considered as potential avenues for future research and not definitive conclusions.

## Conclusion

This scoping review sought to map and describe drug and alcohol-related harm reduction strategies, which have been evaluated in inpatient settings and EDs, the outcome measures used to evaluate these strategies, and implementation characteristics. We identified several gaps in the types and targets of potentially beneficial strategies, outcome measures, and factors related to the implementation of harm reduction strategies for PWUAD. Patient partners provided valuable insight throughout the review process to enrich study findings. The findings of this review may inform future research and will serve as a resource for harm reduction evaluation and implementation efforts in the context of EDs and inpatient settings.

## Supporting information

**S1 Table. Search strategy.**
(XLSX)

**S2 Table. The Preferred Reporting Items for Systematic Reviews and Meta-analyses for Scoping Reviews checklist.** JBI = Joanna Briggs Institute; PRISMA-ScR = Preferred Reporting Items for Systematic reviews and Meta-Analyses extension for Scoping Reviews.* Where sources of evidence (see second footnote) are compiled from, such as bibliographic databases, social media platforms, and Web sites.† A more inclusive/heterogeneous term used to account for the different types of evidence or data sources (e.g., quantitative and/or qualitative research, expert opinion, and policy documents) that may be eligible in a scoping review as opposed to only studies. This is not to be confused with information sources (see first footnote). ‡ The frameworks by Arksey and O'Malley (6) and Levac and colleagues (7) and the JBI guidance (4, 5) refer to the process of data extraction in a scoping review as data charting.§ The process of systematically examining research evidence to assess its validity, results, and relevance before using it to inform a decision. This term is used for items 12 and 19 instead of "risk of bias" (which is more applicable to systematic reviews of interventions) to include and acknowledge the various sources of evidence that may be used in a scoping review (e.g., quantitative and/or qualitative research, expert opinion, and policy document) [19].
(PDF)

**S3 Table. The Guidance for Reporting Involvement of Patients and the Public revised short form (GRIPP2-SF) checklist [21].**
(DOCX)

## Acknowledgments

We would like to acknowledge the support of our research team, specifically Sharon Amey for her work on this project. We would like to thank our partners, Morgan Joudrey, Amanda Hudson-Frigault, Lesley Huska and Caroline Jose. We would also like to acknowledge the

Nova Scotia Health Authority, IWK Health, St Michael's Hospital, the Toronto Opioid Action Network Implementation Committee and the Strategy for Patient-Oriented Research Evidence Alliance for their support of this project.

## Author Contributions

**Conceptualization:** Janet Curran, Mari Somerville, Shannon MacPhee, Annette Elliott Rose, Alexander Caudrella.

**Formal analysis:** Daniel Crowther, Janet Curran, Mari Somerville, Doug Sinclair, Lori Wozney, Shannon MacPhee, Annette Elliott Rose, Alexander Caudrella.

**Funding acquisition:** Janet Curran.

**Investigation:** Janet Curran, Alexander Caudrella.

**Methodology:** Daniel Crowther, Janet Curran, Mari Somerville, Lori Wozney, Leah Boulos.

**Project administration:** Daniel Crowther.

**Resources:** Leah Boulos.

**Supervision:** Janet Curran.

**Writing – original draft:** Daniel Crowther, Mari Somerville.

**Writing – review & editing:** Daniel Crowther, Janet Curran, Mari Somerville, Doug Sinclair, Lori Wozney, Shannon MacPhee, Annette Elliott Rose, Alexander Caudrella.

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
