## [Decision Letter · Decision Letter 0]

27 Feb 2023

PONE-D-22-25311Harm reduction strategies in acute care for people who use alcohol and/or drugs: A scoping reviewPLOS ONE

Dear Dr. Curran,

Thank you for submitting your manuscript to PLOS ONE. After careful consideration, we feel that it has merit but does not fully meet PLOS ONE’s publication criteria as it currently stands. Therefore, we invite you to submit a revised version of the manuscript that addresses the points raised during the review process.

We look forward to receiving your revised manuscript.

Kind regards,

Saeed Ahmed, MD

Academic Editor

PLOS ONE

Reviewers' comments:

Reviewer's Responses to Questions

**Comments to the Author**

1. Is the manuscript technically sound, and do the data support the conclusions?

Reviewer #1: Yes

Reviewer #2: Yes

Reviewer #3: Yes

Reviewer #4: Yes

2. Has the statistical analysis been performed appropriately and rigorously? 

Reviewer #1: Yes

Reviewer #2: Yes

Reviewer #3: Yes

Reviewer #4: I Don't Know

3. Have the authors made all data underlying the findings in their manuscript fully available?

Reviewer #1: Yes

Reviewer #2: Yes

Reviewer #3: Yes

Reviewer #4: Yes

4. Is the manuscript presented in an intelligible fashion and written in standard English?

Reviewer #1: Yes

Reviewer #2: Yes

Reviewer #3: Yes

Reviewer #4: Yes

5. Review Comments to the Author

Reviewer #1: Overall, this is a well-written paper about a topic that needs research. The article starts with a clearly identified and relevant research question. The inclusion and exclusion criteria are clearly written. This study does present results of original research and these results have not been published elsewhere. The article is easy to read and offers actionable knowledge that can be addressed by organizations and healthcare providers. The article does appear to meet standards for ethics and adheres to guidelines and standards for data availability.

Given the ongoing morbidity and mortality caused by substance use, there has been a push to look at novel ways to engage People Who Use Alcohol and/or Drugs (PWUAD). Many of such novel approaches are based in the ER as PWUAD frequently show up there to seek care as they do not have a steady source of income or Insurance (in the USA). Unfortunately, this has also led to some ER physicians and employees developing a stigma against PWUAD. This has been observed by this reviewer on many occasions while talking to MDs not trained in psychiatry. The authors can consider discussing this as a potential motivating factor for healthcare providers reporting a lack of training in addictions as a more palatable reason to refuse providing care. Tackling this stigma is key to increase willingness to implement such strategies mentioned in line 305.

The authors can consider adding a comment about the recent decision by the DEA to drop the X waiver requirement.

The authors can also consider providing housing resources as harm reduction strategies being used in other settings mentioned in line 313.

This article is a great teaching tool to be discussed in schools and residency training programs to ensure healthcare providers are provided education at a stage before they start working as attendings and develop stigma. The authors can consider tabulating the points mentioned from line 354 to 370 as these are great teaching points for a residency program.

Minor concerns

Line 22- please consider rephrasing, the current sentence reads awkwardly.

Line 52- there should be a / in between "and" and "or" like was mentioned in line 21.

Line 55- the comma can be removed.

Line 74 and 75- just because PWUAD presents to the hospital and ED at a higher rate does not mean it is an important point of care. It does make the hospital an ideal location where one can approach PWUAD at a time when they are feeling vulnerable and provide them options. This makes line 78 even more relevant.

Reviewer #2: Reviewer Comments to Author

This is an important and ambitious scoping review on an essential topic. I am glad to see a work so ambitious in manuscript.  Scoping reviews  ideally should systematically identify and chart relevant literature that meet predetermined inclusion criteria. I am glad they grey literature was included as well as stakeholder consultation with PWUD. Hospital based harm reduction strategies can be a critical “touchpoint” for improving the health of people who use drugs. I congratulate them on this big accomplishment getting this done and it will be an important piece for harm reduction in healthcare settings as resonates with me as a clinician and a researcher.

The review seeks to outline four major questions and evaluate the literature in this topic area:

1. What harm reduction strategies have been evaluated to help alleviate negative health outcomes associated with substance use within inpatient settings and EDs?

2. What are the commonly reported outcome measures used to evaluate harm reduction strategies and their implementation in these settings?

3. How are harm reduction strategies implemented in inpatient settings and EDs?

4. What are the reported barriers and enablers to their implementation?

Specific Comments/clarifications requested

1.     The inclusion of alcohol interventions is an interesting one. People who use drugs and alcohol as a term is not well known/used, and the inclusion of people who use alcohol certainly can skew/broaden the patient population since the prevalence of alcohol is high. I think the article could do with clarification about the use of this novel term and justifications as to why. Alcohol harm reduction strategies are a relatively newer field and the inclusion of alcohol harm reduction (managed alcohol, BNI, etc) should be explicated.

2.     I am glad that the study interventions were coded alongside the Hawk principles of harm reduction in healthcare. Certainly some here but appreciate the delineation of the coding process. With some of the harm reduction principle codes, I wonder about the results. i.e. Wakeman et al 2017 multidisciplinary team offering motivational counseling, pharmacotherapy, is that not an example of humanism? Does Dong et al 2020 supervised consumption service intervention not epitomize the principle of autonomy? Etc

3.     Clarifications: Page 16 table 2 – what is Nav Star ? Page 10, Table 1. Welch et al. What is “Relay program” please explicate for a general audience

4.     Clarification: Was any of the grey literature included in the final sample?

5.     Re Table 4: Barriers and Enablers to Implementation

I found this table difficult to read. There were so many barriers, particularly I the “system and organizational factors.” Could this not be further delineated? It might also be useful to identify any barriers and enablers on the same row, i.e. lack of resources, reported enablers “securing funding” if possible. Many of the system and organization factors I think could be further coded and organized.

6.     Discussion could use some re-work and re-organization. Might consider putting partner reflections before the discussion section. Good section on the summary of the key findings in the study, as well as gaps in comparison to the current harm reduction strategies and or previous literature, but would benefit from further explication of research strategies, areas for more rigorous investigation including methods, as well as clinical, policy and research implications. It would be notable to also mention the distinct policy implications of implementation that are not generalizable across the countries included in this scoping review.  

7.     I feel the partner/stakeholder reflections section is also incomplete. Were these direct quotes from participants at CBOs and or PWUD? They are listed out almost like direct quotes rather than summarized. Can they be summarized or organized differently? Line 364: “often feel like they must police their own treatment” (?) unclear what they mean by this. Reads very disorganized/unclear, not as cohesive as the rest of the manuscript

Reviewer #3: Congratulation to the authors for your work on an important topic. The manuscript draws up a comprehensive database search to find research papers. However, the manuscript rests on many case reports. Moreover, studies from more than one country are included. This is important because one of the identified factors is organizational limitations. Laws, regulations, organizational structure, work culture, funding, etc., tend to differ in different countries. Also, organization limitation is a vague term and has yet to be defined by the authors. Illicit substance-induced disorders like opiates, amphetamines, marijuana, and alcohol abuse disorders share many socio-economic challenges. Still, many factors may vary, which can impact harm reduction strategies for each illicit substance, such as the level of stigma and the general approach of clinicians toward patients with amphetamine will be different from marijuana. Also, laws about illicit substances vary per country. The laws about other illicit substances within a country can vary, e.g., marijuana.

Reviewer #4: Excellent work with this review. You have done remarkable work with this article. Though not easy to conduct study like this you have done phenomeninal work with this.

---

## [Author Response · Author response to Decision Letter 0]

3 Aug 2023

Authors’ Response

Thank you for this comment. We agree that stigma toward PWUAD by providers is a complex issue which should be addressed. We have highlighted the potential for strategies, such as training, in EDs on line 359-362

A comment about the X-waiver decision has been added to the introduction (page 3, line 74-77)

This sentence was amended to include a reference related to providing housing as a harm reduction strategy (page 24, line 366)

To improve readability the Partner Reflections on Gaps in the Literature section has been placed in a table and has been moved before the discussion. (page 23, line 325-326)

This sentenced has been amended for clarity (page 2, line 21-23)

A “/” has been added between the “and” and “or” (page 3, line 57)

This comma has been removed (page 3, line 59)

Line 74 and 75 have been amended and the sentence now states that the hospital is an ideal point of care and not an important point of care. (page 3, line 82)

Authors’ Response

Thank you for your comment. Based on our review people who use drugs and/or alcohol is not a novel or uncommonly used term. It is common in the field of harm reduction research. Literature using this or similar terms date to at least 2004 and this term has been in use as of 2022. Please refer to this selection of literature for reference:

– https://doi.org/10.1177/02692163211061994

– https://doi.org/10.1186/s12954-020-00366-3

– DOI 10.1007/s10461-013-0587-9 

– http://www.biomedcentral.com/1471-2458/12/312

– https://doi.org/10.1089/10872910050046340

Harm reduction strategies for alcohol have been in use and researched as early as 1983. Literature related to this topic also continues to be published. Please refer to this selection of literature for reference:

– DOI: 10.1080/09595230600944529

– https://doi.org/10.1016/j.drugpo.2006.05.004

– https://doi.org/10.1037/a0026397

– https://doi.org/10.3389/fpubh.2022.855416

We appreciate that textual coding involves a certain level of subjective interpretation. Our research team attempted to enhance the rigor of our coding by having two reviewers independently code the included studies based on Hawk’s principles. Discrepancies between codes were brought to the team, who discussed the studies and assessed which code was appropriate based on their knowledge of the included studies and their expertise. However, we acknowledge that this process is not completely objective, and this has been noted as a limitation within the limitation section. (page 25, line 412-414)

The authors have clarified that the Relay Program (Table 1) and NavStar (Table 2) are the names of harm reduction strategies. The content of these strategies is described in the “Intervention Content” column in Table 2 (page 14 and 16).

No grey literature was included in the final sample. (page 6, line 194-195, and Figure 1).

The table has been reformatted for clarity and readability. (page 20-22, Table 4)

We have added further description in the Data synthesis and presentation section within the Methods to clarify meaning related types of barriers and enablers. (page 6, line 166-173) 

To improve clarity and readability the Partner reflections have been tabulated and moved before the discussion section. (page 23, Table 5)

As mentioned above, the Partner reflections have been tabulated and moved before the discussion section to improve clarity and readability. 

The partner reflections are not direct quotes, but a summary of reflections of our partners. The table has been renamed and several of the statements have been reworded to clarify this. (page 23, Table 5)

Authors’ Response

We agree our comprehensive search is a strength of this review. We would agree that organizational limitation is a vague term, and we would like to note that it was not used as a term in this manuscript or any of the included studies. As mentioned above, we amended the Data synthesis and presentation section within the Methods to include a definition of the term system level and organizational factors. (page 6, line 166-173) 

We agree with the reviewer that the quality of a study’s findings is dependent on its study design and that barriers and enablers to implementation may vary depending on the country where implementation takes place. However, we would like to emphasize that this review provides a summary of the scope of the available literature related to our topic which was captured by our search strategy and our inclusion criteria. This review is not a meta-analysis, and we are not assessing the generalizability of the results of the included papers or the extent to which study design may limit the generalizability of the findings within each study. To clarify this, the limitation section has been amended with a sentence to that effect. (page 25, line 411-412)

---

## [Decision Letter · Decision Letter 1]

10 Nov 2023

Harm reduction strategies in acute care for people who use alcohol and/or drugs: A scoping review

PONE-D-22-25311R1

Dear Dr. Curran,

We’re pleased to inform you that your manuscript has been judged scientifically suitable for publication and will be formally accepted for publication once it meets all outstanding technical requirements.

Kind regards,

Ricky N. Bluthenthal

Academic Editor

PLOS ONE

Additional Editor Comments (optional):

Congratulations on a thorough and important contribution to examining the use of harm reduction strategies in acute care settings. I think the paper should just use the definition of harm reduction described by Hawk as suggested by reviewer 1. I also agree that methadone is not traditionally considered an harm reduction approach (although it is largely used by one many people on methadone). Addressing these two comments and the few other issues identified by reviewers would improve the paper. Also, this review process has taken too long and I want to apologize for this.

Reviewers' comments:

Reviewer's Responses to Questions

**Comments to the Author**

1. If the authors have adequately addressed your comments raised in a previous round of review and you feel that this manuscript is now acceptable for publication, you may indicate that here to bypass the “Comments to the Author” section, enter your conflict of interest statement in the “Confidential to Editor” section, and submit your "Accept" recommendation.

Reviewer #2: (No Response)

Reviewer #3: All comments have been addressed

2. Is the manuscript technically sound, and do the data support the conclusions?

Reviewer #2: Yes

Reviewer #3: Yes

3. Has the statistical analysis been performed appropriately and rigorously? 

Reviewer #2: Yes

Reviewer #3: Yes

4. Have the authors made all data underlying the findings in their manuscript fully available?

Reviewer #2: Yes

Reviewer #3: Yes

5. Is the manuscript presented in an intelligible fashion and written in standard English?

Reviewer #2: Yes

Reviewer #3: Yes

6. Review Comments to the Author

Reviewer #2: Overall this is a significantly improved manuscript and the authors have done well to improve readability and clarity for a paper on a very important subject. I think this is reaching a level where revisions will be appropriate before publication in this journal.

A couple points:

1. Line 65: “Harm reduction is an approach that emphasizes working with people where they are at, rather than focusing solely on drug and alcohol abstinence”. The citation (7) used a definition of harm reduction that said “A policy, programme or intervention should only be called harm reduction if, and only if: (1) the primary goal is the reduction of drug-related harm rather than drug use per se; (2) where abstinence-orientatated strategies are included, strategies are also included to reduce the harm for those who continue to use drugs; and (3) strategies are included which aim to demonstrate that, on the balance of probabilities, it is likely to result in a net reduction of drug-related harm.” Some other definitions of harm reduction might be utilized here for further clarity rather than working with people where they are at, which is vague. Suggestions:

Harm reduction refers to interventions aimed at reducing the negative effects of health behaviors without necessarily extinguishing the problematic health behaviors completely (Hawk paper).

The National Harm Reduction Coalition defines harm reduction as “a set of practical strategies and ideas aimed at reducing negative consequences associated with drug use”

2. Line 68: It is controversial to state that MMT is a harm reduction strategy, that might just be considered evidence-based standard of care for treatment of opioid use disorder. Potentially saying low threshold MOUD (?) programs including buprenorphine

3. Line 70-71: Again, not to be pedantic, but removal of X-waiver for use of MOUD is not necessarily a harm reduction strategy but a treatment strategy aimed at expanding medication for OUD which is certainly necessary. US HHS/SAHMSA has put 4 buckets of responses across these categories: prevention, treatment, harm reduction and recovery supports.

4. Characteristics outcome measures: (line 262) I believe that these outcome measures and the organization of outcomes into these categories within harm reduction are a bit shoe-horned and potentially not the right category for analysis. The safer use outcomes seem clear. In my understanding of managed use, in harm reduction, this is a strategy utilized by programs or patients to control and moderate substance use, such as managed alcohol programs during COVID in shelters or housing programs. Therefore, saying that “managed use was compromised of outcome measures related to referral to/and or acceptance of care, satisfaction and/or experience of care and HCP follow-up” seems an inappropriate outcome measure category, which might be better expressed as healthcare utilization and engagement or satisfaction. In addition, conditions of use and use itself to me include harm reduction strategies such as use of sterile syringes, using with naloxone, not using alone, etc. I think stating that “conditions of use and use itself included measures related to mortality, readmission rates, leaving against medical advice, adverse events, length of stay and frequency of drug and/or alcohol” is perhaps a not a good way to categorize these outcomes, which might be other variables of healthcare utilization.

5. Table 4: “difficulty scheduling or locating patients for training” had no citation paper

- What is MAP (56)

- System and Organizational Factors: Reported Enablers of Implementation move to the next page for readability

- Outsourcing SBIR to whom (80)

6. Partner reflections: clarify regarding “reporting of neglect or abuse” (is that by HCP)? Healthcare worker abuse or neglect (?).

- Strategies to deal with adverse events such as allergic reactions (to any medication, not just naloxone or methadone)? Perhaps it is stigmatizing to call these two medications out in particular and should just be left broadly as allergic reactions or adverse side events

Reviewer #3: Authors have addressed prior concerns noted by all reviewers. Line 360, authors should give examples of organizational level factors. Harm reduction strategies are different for alcohol and illicit substances. Also, systemic barriers as well as laws are different in different countries. Perhaps authors can consider addressing same in limitations.

7. PLOS authors have the option to publish the peer review history of their article (what does this mean?). If published, this will include your full peer review and any attached files.

Reviewer #2: No

Reviewer #3: No

---

## [Editor Report · Acceptance letter]

5 Dec 2023

PONE-D-22-25311R1 

Harm reduction strategies in acute care for people who use alcohol and/or drugs: A scoping review 

Dear Dr. Curran:

I'm pleased to inform you that your manuscript has been deemed suitable for publication in PLOS ONE. Congratulations! Your manuscript is now with our production department. 

Kind regards, 

on behalf of

Dr. Ricky N. Bluthenthal 

Academic Editor

PLOS ONE